# A *Bacillus thuringiensis* Cry protein controls soybean cyst nematode in transgenic soybean plants

Theodore W. Kahn [ID] [1✉], Nicholas B. Duck[1,2], Michael T. McCarville[1], Laura Cooper Schouten[1], Kathryn Schweri[1], Jelena Zaitseva [ID] [1] & Julia Daum[1]

Plant-parasitic nematodes (PPNs) are economically important pests of agricultural crops, and soybean cyst nematode (SCN) in particular is responsible for a large amount of damage to soybean. The need for new solutions for controlling SCN is becoming increasingly urgent, due to the slow decline in effectiveness of the widely used native soybean resistance derived from genetic line PI 88788. Thus, developing transgenic traits for controlling SCN is of great interest. Here, we report a *Bacillus thuringiensis* delta-endotoxin, Cry14Ab, that controls SCN in transgenic soybean. Experiments in *C. elegans* suggest the mechanism by which the protein controls nematodes involves damaging the intestine, similar to the mechanism of Cry proteins used to control insects. Plants expressing Cry14Ab show a significant reduction in cyst numbers compared to control plants 30 days after infestation. Field trials also show a reduction in SCN egg counts compared with control plants, demonstrating that this protein has excellent potential to control PPNs in soybean.

[1] BASF, Morrisville, NC, USA. [2]Present address: Avertica, Research Triangle Park, NC, USA. ✉email: ted.kahn@basf.com

Plant-parasitic nematodes (PPNs) are major agricultural pests around the world, resulting in about $80 billion of damage per year[1]. Losses in row crops are mostly due to cyst nematodes (*Heterodera* and *Globodera* spp.), root-knot nematodes (*Meloidogyne* spp.), and lesion nematodes (*Pratylenchus* spp.). Soybean cyst nematode (SCN), *Heterodera glycines*, is the most important pest of soybean in the United States, responsible for more than twice as much yield loss as the next greatest cause[2], and is one of the major pests of soybean in Brazil[3]. *H. glycines* can reduce yield by 30–50%, even in fields with little or no visible above-ground symptoms[4,5]. In the U.S. control of nematodes has relied on the use of varieties incorporating native resistance[6], crop rotation, and nematicidal seed treatments[7]. One source of native resistance, derived from plant introduction (PI) accession 88788, results from a high copy number of the *rhg1-b* allele[8]. PI 88788 is now used in 95–98% of *H. glycines*-resistant soybean varieties grown in the U.S., but its effectiveness against this pest has recently begun to decline[9].

*Bacillus thuringiensis* (*Bt*) is a Gram-positive sporulating bacterial species that produce a wide variety of insecticidal proteins, including the family of Cry (Crystal) proteins. *Bt* proteins are among the most broadly used natural insecticides in agriculture[10]. Crops have been engineered to express *Bt* insecticidal proteins for controlling pests such as Lepidoptera and Coleoptera, generating commercial products that have delivered significant benefits to farmers over the past 25 years[11,12]. *Bt* pesticides have a record of safety in agriculture stretching back more than half a century, with no significant risk to the environment or to human health[13]. Some *Bt* proteins have been shown to be active against nematodes[14,15], but to date, there have been no transgenic crops on the market that control nematodes.

*H. glycines* is an obligate parasite that cannot easily be maintained or fed test substances in vitro[16]. *Caenorhabditis elegans* has the advantage that it can easily be grown in vitro, and can be fed bacteria that express potential protein toxins, and can also be fed purified proteins. Genetically *C. elegans* has much in common with PPNs[17], although it lacks genes involved in parasitism[18]. For these reasons, *C. elegans* is a useful surrogate for identifying proteins that will damage the nematode digestive system[19], and

can potentially be expressed in transgenic crops to control nematodes in the same way that orally active insecticidal proteins have been used to control insect pests.

In this study, we describe a nematode-active *Bt* Cry protein, Cry14Ab, that is highly active against *C. elegans*. We show that Cry14Ab is somewhat unusual compared to many insecticidal Cry proteins in that the C-terminal half of the protein, commonly referred to as the crystallization region[20], is not easily removed in vitro by common proteases[21]. However, in other ways, Cry14Ab is typical of Cry proteins in that it associates with the nematode intestine and causes damage to the intestine, consistent with the mechanism of action of insecticidal Cry proteins. Transgenic soybean events expressing Cry14Ab are protected from *H. glycines* in the greenhouse and field. The expression of the protein and the resistance to *H. glycines* is stable in F2 generations of the transgenic soybean plants, showing that Cry14Ab is a good candidate for commercial control of *H. glycines* in soybean.

## Results

**Discovery of the strain.** Bacterial strains were isolated from soil and tested individually for activity against *C. elegans* in a liquid bioassay. *C. elegans* consumes whole bacteria, and will also consume soluble proteins[22]. The bioassays included a laboratory *E. coli* strain (HB101) as worm food, as well as the bacteria or other material being tested for activity. A protein toxin need not be soluble in order to be active in this bioassay. Approximately 50 worms were placed in each well of a multi-well plate along with the material being tested for activity. Three to 5 days later, after the nematodes have had time to grow and reproduce, the wells were scored by eye under a dissecting microscope for the approximate number of nematodes present, the level of mobility of the nematodes, and the amount of clearing of worm food bacteria from the well. As shown in Fig. 1, a visual score from 0 to 5 was assigned to each well. A score of 0 was assigned to wells that resembled negative control wells, with hundreds of very active nematodes and almost complete clearing of worm food bacteria, indicating that the test material had no toxic effect on the nematodes. Higher score numbers were assigned when fewer and

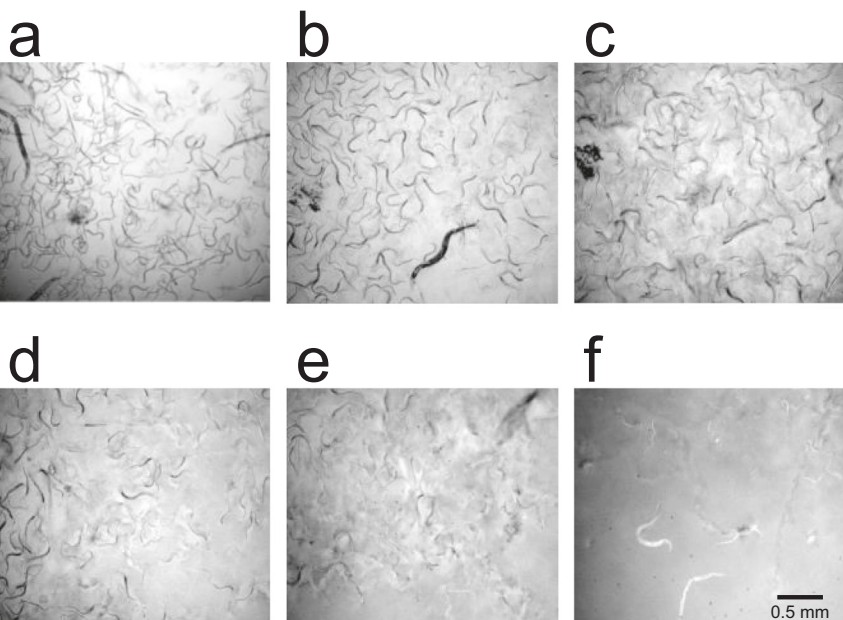

**Fig. 1 Micrographs of *Caenorhabditis elegans* in liquid bioassay wells, illustrating the visual scoring system.** Panels **a** through **f** show representative images of wells with scores of 0 through 5, respectively. This scoring system was developed based on thousands of observations.

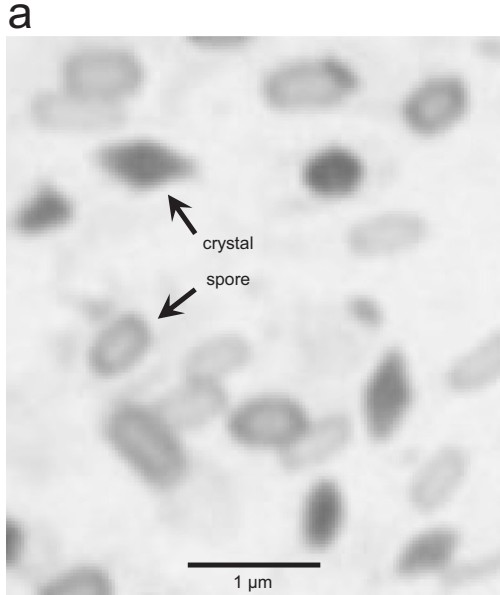

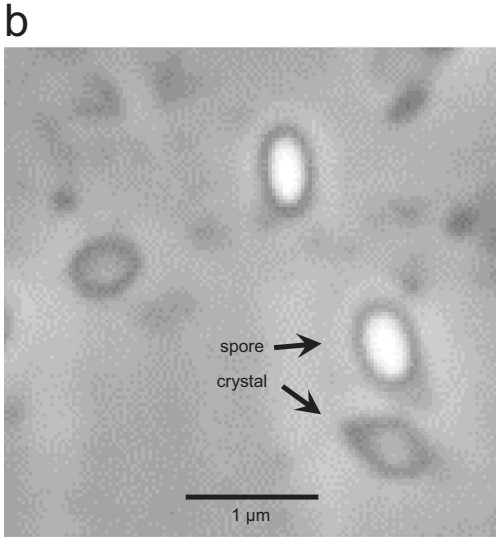

**Fig. 2 Micrographs of bacterial cultures, taken with a ×100 oil immersion lens.** Cultures were grown and observed microscopically five times each, and similar results were seen each time. Photographs were taken once. **a** Sporulated native *Bacillus thuringiensis* (*Bt*) strain after autolysis, stained with Coomassie. Crystals are stained dark, while spores remain light. **b** Sporulated plasmid-cured *Bt* strain expressing Cry14Ab, after autolysis. Source data are provied as a Source Data file.

less active nematodes were observed, and when cloudiness from unconsumed worm food bacteria was present, indicating progressively stronger toxic effects of test substances. A score of 5 was assigned when only a few, sluggish worms were present, and the well was cloudy with unconsumed worm food bacteria, showing that the test material had inhibited the survival and reproduction of the nematodes.

Using this bioassay method, an active bacterial strain was identified. Bioassay scores of 3–5 were seen when this strain was grown for 1–5 days before being added to the bioassays. This bacterial strain was incapable of growing on nutrient agar containing a combination of tetracycline, chloramphenicol, and cefotaxime. When that combination of antibiotics was added to

the *C. elegans* bioassays the results were the same as in the absence of the antibiotics, suggesting that activity against the nematodes may not have been dependent on active strain growth, and encouraging us to investigate the possibility that the strain may have released an active substance into the bacterial culture medium.

**Characterization of activity**. Cultures of the active bacterial strain were centrifuged to separate them into soluble and insoluble fractions. The soluble fractions were passed through 0.2 μm filters to remove any remaining cells, spores, crystals, or other insoluble material. The cell pellets were resuspended at the original concentration. The cell pellets showed activity (scores of 3–5) in bioassays. The sterile-filtered fractions were active when prepared from large cultures (30 mL) but not when prepared from small cultures (1 or 2 mL), showing that the cultures always contained insoluble active material, and sometimes contained soluble active material. When the insoluble pellet was extracted at pH 10 with dithiothreitol, the extract was active after being sterile filtered, showing that high pH could solubilize the insoluble active material. Spin filters with a 3 kDa molecular weight cutoff retained activity from sterile-filtered soluble samples, showing that the active moiety was larger than 3 kDa. No activity was seen (score of 0) when either the whole cultures, resuspended cultures, sterile-filtered culture supernatants, or sterile-filtered high pH extracts were heated to 95 °C for 10 min before being added to the bioassays, showing that the activity was heat-sensitive, and suggesting the possibility that a protein was responsible for the activity.

**Identification of strain**. The active strain was initially indicated to be *Bacillus cereus* by MIDI (The Sherlock™ Microbial Identification System) fatty acid analysis. DNA sequencing later showed the strain to be *B. thuringiensis*, with the taxonomic assignment being based on a whole-genome K-mer spectrum comparison to the NCBI RefSeq database, on 16S sequence, and on total genome average nucleotide identity compared to RefSeq. Microscopic observation revealed that the strain produces crystals (see Fig. 2a), which also indicated that the strain was *Bt*.

**Identification of active molecule**. To identify the molecule responsible for nematicidal activity in the *Bt* strain, a 100 mL culture was grown to sporulation and autolysis, the insoluble material was pelleted, and the supernatant was passed through a 0.2 μm filter to sterilize it. The sterile culture supernatant was fractionated by anion exchange chromatography, and the fractions were tested for activity in bioassay. Treatment of the active fractions with immobilized trypsin for 4 h at 37 °C did not reduce the activity. The trypsin reduced the intensity of many of the protein bands in the active fractions, but not of a band at ~130 kDa. One active fraction was concentrated, run on sodium dodecyl sulfate–polyacrylamide gel electrophoresis (SDS–PAGE), blotted to polyvinylidene fluoride (PVDF), and stained with Coomassie. Bands from the gel were subjected to N-terminal sequencing (Supplementary Fig. 1).

The N-terminal sequences of the 130 and 70 kDa bands matched the N-terminus of Cry14Aa (GenBank accession AAA21516), suggesting they represented a full-length and truncated version of a protein similar to Cry14Aa, which may have been partially cut by an endogenous protease in the bacterial strain at approximately amino acid position 630–640. Cry proteins are typically produced by *Bt* in the form of insoluble crystals that can only be solubilized at very high or very low pH[23], but in this case, enough of the protein was secreted by the bacteria in soluble form into the culture medium at ~pH 8 that we were

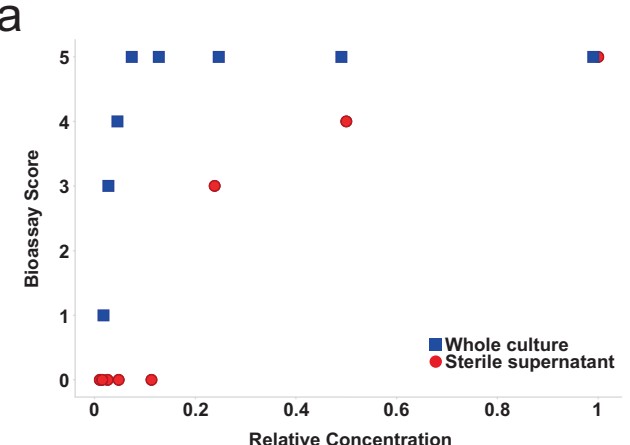

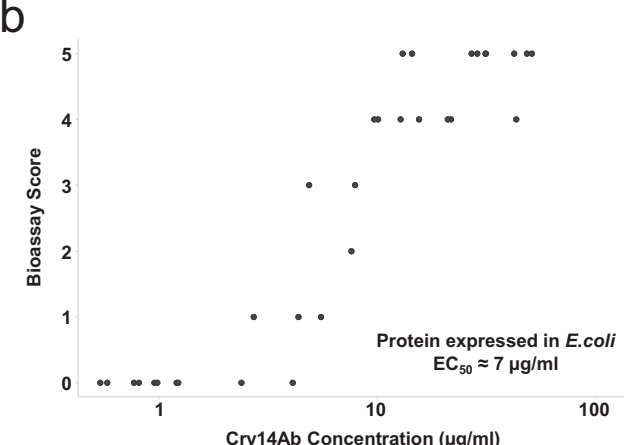

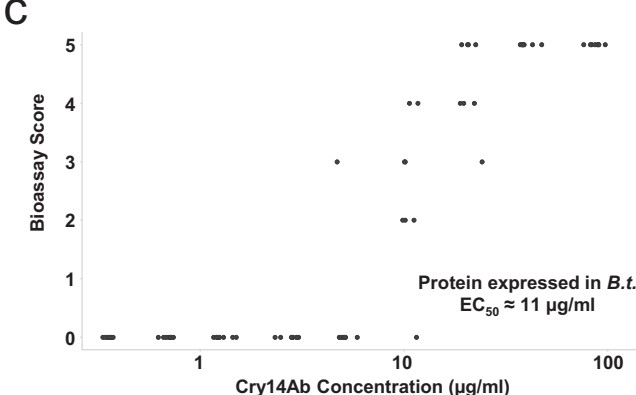

**Fig. 3 Effect of Cry14Ab on *Caenorhabditis elegans* in liquid bioassays.** The bioassay visual scoring system is explained in Fig. 1. Graphs were prepared using TIBCO Spotfire, version 7.10.1. **a** Whole culture (blue squares) or sterile supernatant (red circles) samples from a culture of the native *Bt* strain carrying the Cry14Ab gene were fed at different dilutions, with a relative concentration of 1 indicating no dilution. Each data point represents a single bioassay measurement. The symbols have been jittered in the horizontal direction to make them more visible. **b** Purified Cry14Ab expressed in *Escherichia coli* was fed at different concentrations. Two technical replicates were collected at each concentration. The symbols have been jittered in the horizontal direction to make them more visible. **c** Purified Cry14Ab expressed heterologously in a plasmid-cured *Bt* strain was fed at different concentrations. Eight technical replicates were collected at each concentration. The symbols have been jittered in the horizontal direction to make them more visible. Source data are provided as a Source Data file.

assigned the name Cry14Ab by the Bacterial Pesticidal Protein Resource Center (https://www.bpprc.org/).

**Heterologous expression in bacteria.** The full-length PCR product of the Cry14Ab gene was inserted into a vector for expression in *E. coli* with an N-terminal 6his-maltose-binding protein tag, and into a vector for expression in a plasmid-cured strain of *Bt*[24] with no tag. In *E. coli* the fusion protein was soluble at pH 8 and pH 10.5. In *Bt* the protein formed crystals with a similar appearance to the crystals in the native strain (Fig. 2b, above), but some soluble protein was also present in the culture supernatant, as was the case with the native strain. This similarity leads us to suspect that the crystals produced by the native strain also contain Cry14Ab, although we did not analyze those crystals.

**$EC_{50}$ of Cry14Ab.** The activity level of the native strain carrying the Cry14Ab gene, and of the purified protein itself, were measured using the liquid *C. elegans* bioassay described above. A dilution series of the native *Bt* strain carrying the Cry14Ab gene was fed to *C. elegans* as both the whole bacterial culture and the sterile-filtered culture supernatant (Fig. 3a). Both showed activity, indicating that the active species is partially soluble under the bacterial culturing conditions (~pH 8). Dilution caused the activity of the sterile supernatant to drop faster than the activity of the whole culture, showing that a significant amount of the active moiety is present in an insoluble form, perhaps as crystals still contained within intact bacterial cells or released by autolysed cells.

A dilution series of purified Cry14Ab expressed heterologously in *E. coli* as a fusion with maltose-binding protein (Fig. 3b), or untagged in an acrystalliferous (plasmid-cured) *Bt* strain (Fig. 3c), was fed to *C. elegans*, and the approximate effective concentration giving a response halfway between a score of 0 and 5 ($EC_{50}$) was calculated[25]. Protein from the two sources gave approximately the same $EC_{50}$, with a value of about 7 µg/mL for the protein expressed in *E. coli* (taking into account only the mass of the Cry14Ab portion of the fusion protein) and about 11 µg/mL for the protein expressed heterologously in *Bt* Other nematicidal toxins have roughly similar levels of activity against *C. elegans*, including Cry14Aa (16 µg/mL), Cry21Aa (47 µg/mL), and Cry5Ba (66 µg/mL)[19].

**Cry14Ab is not easily processed to a protease-resistant core.** Cry proteins that are toxic to insects are typically solubilized in the insect gut and then cleaved by gut proteases to a protease-resistant core. In lepidopteran insects, trypsin-like or

able to purify and identify the protein without making use of crystals. Ultracentrifugation at 280,000 × *g* for 2 h pelleted the soluble active material, showing that the protein may be present as large oligomers. Based on the similarity to Cry14Aa, degenerate polymerase chain reaction (PCR) primers were designed (see Supplementary Table 1, primers 050324C and 050324B), and the gene was cloned and sequenced.

**Description of Cry14Ab.** The protein has 1185 amino acids, with a molecular weight of 132 kDa (GenBank: AGU13817.1). Based on sequence alignment it is a 3-domain delta-endotoxin, and it falls into the phylogenetic family of Cry toxins known to be active against *C. elegans*, including Cry5, Cry14, and Cry21[19]. Specifically, it is 87% identical to Cry14Aa at the amino acid level. It was

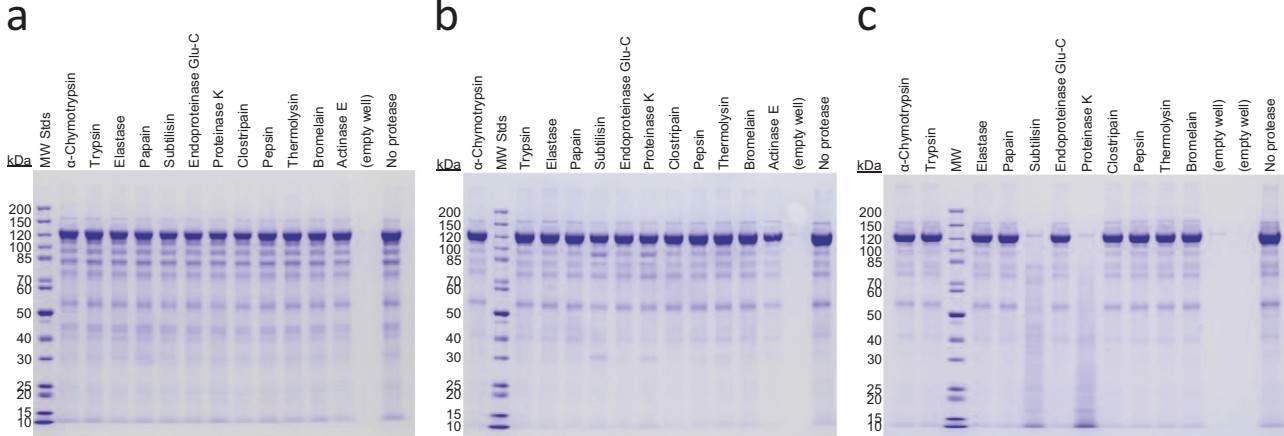

**Fig. 4 Treatment of Cry14Ab with proteases.** Cry14Ab was incubated with various proteases at a protease-to-protein weight ratio of 1:100 at 37 °C for 2 h at three pH values, and then run on SDS–PAGE and stained with Coomassie blue dye. The digestion was performed three times and gave similar results each time. **a** 20 mM sodium acetate pH 5.0, 150 mM NaCl; **b** 5 mM HEPES (4-(2-hydroxyethyl)-1-piperazineethanesulfonic acid) pH 7.5, 250 mM NaCl; **c** 20 mM CAPS (N-cyclohexyl-3-aminopropanesulfonic acid) pH 10.4, 150 mM NaCl.

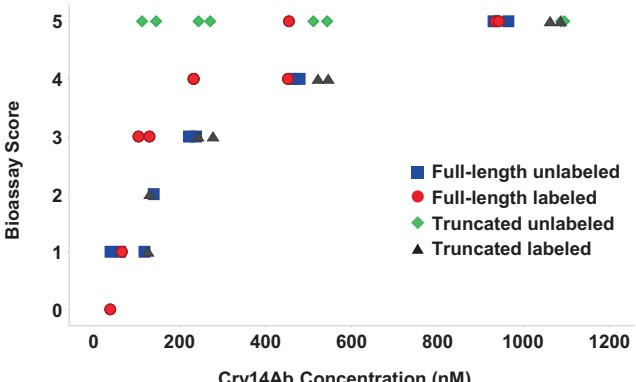

**Fig. 5 Effect of full-length versus truncated, and fluorescently labeled versus unlabeled, Cry14Ab on *Caenorhabditis elegans* in liquid bioassays.** Full-length protein was wild type, while truncated protein had the mutation G708R, and was cleaved at that position with trypsin, followed by purification of the N-terminal fragment. Two technical replicates of each sample were assayed. The symbols have been jittered in the horizontal direction to make them more visible. The graphs were prepared using TIBCO Spotfire, version 7.10.1. Blue squares: full-length unlabeled. Red circles: full-length labeled. Green diamonds: truncated unlabeled. Black triangles: truncated labeled. Source data are provided as a Source Data file.

chymotrypsin-like proteases cleave 130 kDa Cry toxins to about half their original size, removing the C-terminal crystallization portion of the protein from the toxic N-terminal portion[26]. Cleavage typically occurs at a trypsin or chymotrypsin cleavage site located shortly after an amino acid sequence similar to DRIEF. In Cry14Ab no apparent protease cleavage site is located in that region (-DRIEFVPTMPVPGNTNNNNGNNNGNNNPP HH-). To test whether Cry14Ab is susceptible to the typical type of proteolysis seen with Cry proteins, it was incubated with 11 different proteases: α-chymotrypsin, trypsin, elastase, papain, subtilisin, endoproteinase Glu-C, proteinase K, clostripain, thermolysin, bromelain, and actinase E, at pH 5.0, pH 7.5, and pH 10.4. Note that although pepsin was tested in these experiments, it is not active at the pH values used here. As seen in Fig. 4, subtilisin and proteinase K largely destroyed the protein at pH 10.4, while the other proteases had no effect under the conditions used. This shows that Cry14Ab is not easily cleaved to a protease-resistant 55–65 kDa core the way many insecticidal Cry toxins

typically are[27], although it remains possible that processing conditions we did not explore exist in the *C. elegans* or *H. glycines* gut.

When the amino acid at position 708 was mutagenized from glycine to arginine, the protein could be easily cleaved by trypsin. A single *C. elegans* bioassay was done on a dilution series of the cleaved protein (the experiment was not repeated, and precise $EC_{50}$ values were not determined), and the results showed that in *C. elegans* the C-terminal region of the protein is not required for activity (see Fig. 5, full-length unlabeled and truncated unlabeled).

**Effect of Cry14Ab on the intestine of *C. elegans*.** To observe the effect of Cry14Ab on the intestine of *C. elegans*, nematodes were fed fluorescently labeled full-length Cry14Ab (132 kDa), fluorescently labeled truncated Cry14Ab (77 kDa), or fluorescently labeled bovine serum albumin (BSA; 66 kDa) for 24 h. The fluorescent label allowed us to ensure that the nematodes had actually ingested the protein. As a first step, we determined that the labeling had not destroyed the activity of Cry14Ab by performing a single *C. elegans* bioassay on a dilution series of labeled full-length and truncated Cry14Ab (the experiment was not repeated, and precise $EC_{50}$ values were not determined). The results showed that labeled Cry14Ab had about the same level of toxicity as unlabeled protein (see Fig. 5, full-length unlabeled and full-length labeled), while BSA was not toxic. We then fed the labeled proteins to nematodes and observed the worms by microscopy (Fig. 6a through c). Under white light, the BSA-fed worms appeared normal, similar to worms that were incubated for the same length of time in S media without any added test material. By contrast, Cry14Ab-fed worms showed damage to the intestine. Similar damage was seen when the worms were treated with the purified N-terminal region of the trypsin-truncated protein with the G708R mutation, showing that full-length protein is not required for the damage to occur. This type of damage is similar to that seen when Cry5Ba, Cry14Aa, or Cry21Aa is fed to *C. elegans*[19] and suggests that Cry14Ab kills *C. elegans* by damaging the intestine, which is also the mechanism by which other Cry toxins kill insects[28].

To demonstrate that the labeled protein was actually being ingested, worms were fed labeled protein for 4 h, rinsed to remove excess labeled protein, and imaged by fluorescence microscopy. In the case of both labeled Cry14Ab and labeled BSA, the lumen of the intestine was seen to be filled with labeled protein (Fig. 6d).

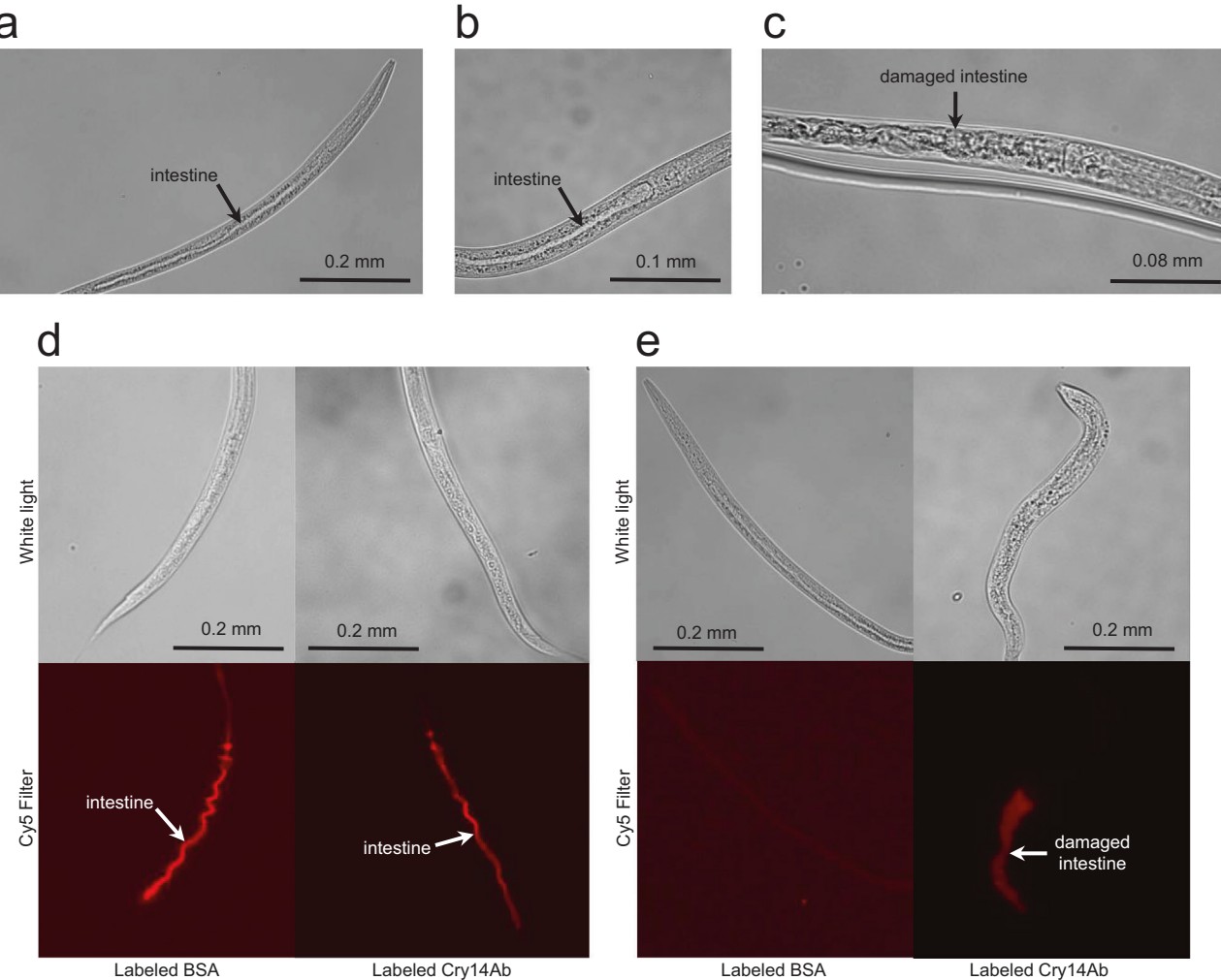

**Fig. 6 Bovine serum albumin and Cry14Ab were fluorescently labeled and fed to *C. elegans* to observe the effects on the intestine.** The experiment was performed twice. Under each condition 50 nematodes were observed, 3 were photographed, and 1 is shown here. **a** Negative control, with no fed protein, after incubation for 24 h; **b** labeled BSA fed for 24 h; **c** labeled Cry14Ab fed for 24 h, showing damage to the intestine. **d** To demonstrate that labeled protein was being ingested, worms were fed labeled protein for 4 h, washed to remove excess labeled protein, and imaged under white light or with a Cy5 filter set. **e** To determine if Cry14Ab remained associated with the intestine, worms were fed labeled protein for 24 h, washed to removed excess labeled protein, and incubated without labeled protein for another 24 h before being imaged under white light or with a Cy5 filter set. Source data are provided as a Source Data file.

To determine if Cry14Ab was being retained in the intestine, ~50 worms were fed labeled protein for 24 h, rinsed to remove excess labeled protein, and placed in a medium without labeled protein for another 24 h, after which about 75% of the worms were still alive. Worms that had been fed labeled BSA showed no fluorescence in the intestine, but all surviving worms that had been fed labeled Cry14Ab retained some fluorescence, showing that the protein remained associated with the intestine (Fig. 6e). Similar results were seen when the worms were fed fluorescently labeled purified N-terminal region of trypsin-truncated G708R protein, showing that the C-terminal portion of the protein is not needed for association with intestinal cells in *C. elegans*. The retention of Cry14Ab is consistent with the fact that 3-domain delta-endotoxins bind to receptors on the surface of intestinal cells[29].

**Induced ingestion of Cry14Ab by *H. glycines*.** At this point, we switched to working with *H. glycines*, the pest of interest. *H. glycines* is an obligate parasite that does not normally feed until it has established specialized feeding sites within roots[16], but newly

hatched second-stage juveniles (J2) that have not yet entered roots can be induced to ingest substances dissolved in a liquid by adding the neurotransmitter octopamine to the liquid[30]. To determine if J2 *H. glycines* nematodes could ingest a 132 kDa protein, fluorescently labeled full-length Cry14Ab at a concentration of 1 mg/mL was fed to the nematodes in liquid, using octopamine to stimulate ingestion. The fluorescently labeled protein was seen in the esophagus and intestine, showing that a protein of this size can pass through the nematode's stylet (Fig. 7a). No fluorescence was observed in nematodes that had not been fed fluorescently labeled protein (Fig. 7b).

**Greenhouse evaluation of efficacy of Cry14Ab against *H. glycines* in soybean.** Soybean cv. Jack was transformed with Cry14Ab. Jack, which was released in 1989 as an *H. glycines*-resistant variety[31], carries a high copy number of the *rhg1-b* allele from PI 88788. Jack is not currently used commercially because of its relatively low yield, but it was chosen for this study because plants can easily be regenerated after being transformed with foreign genes using the aerosol beam transformation method[32].

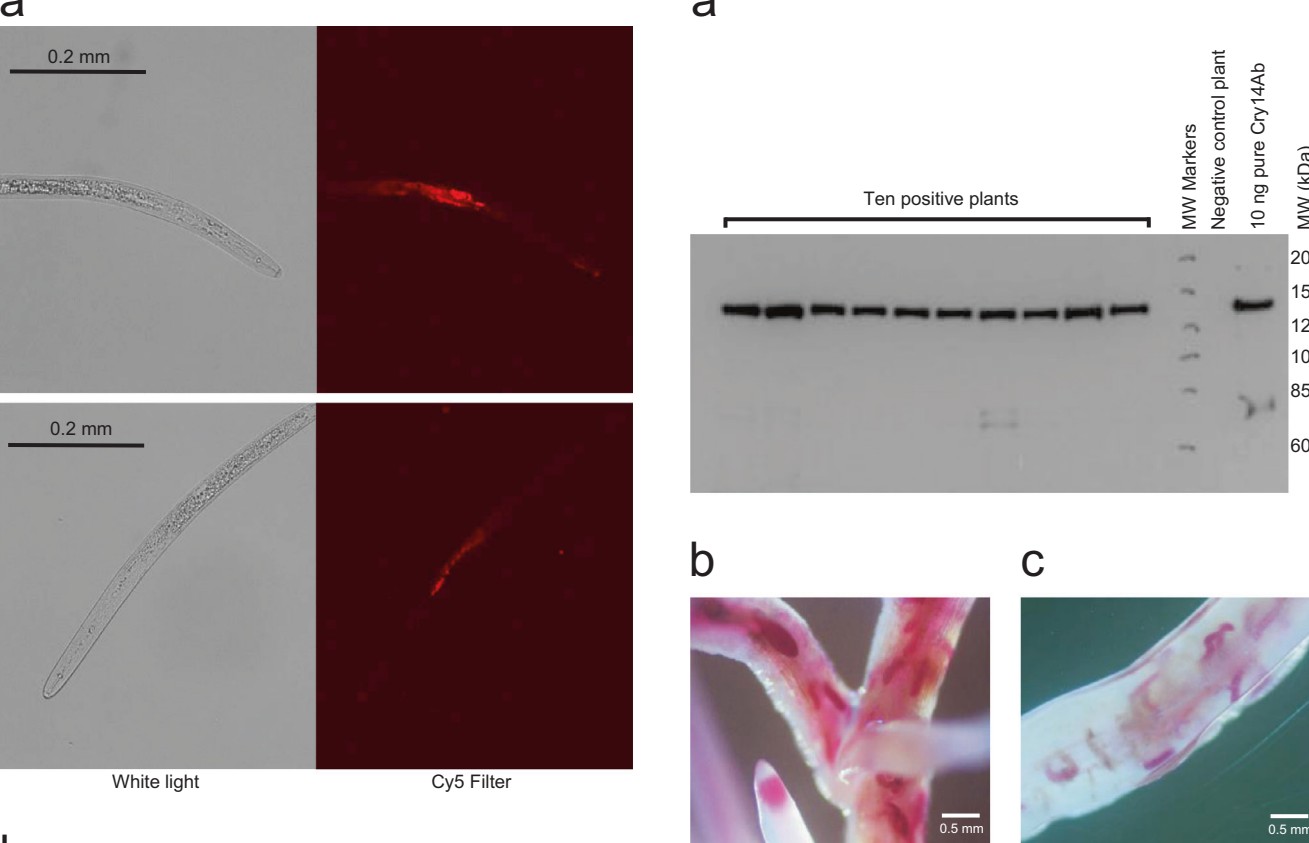

**a**

Ten positive plants

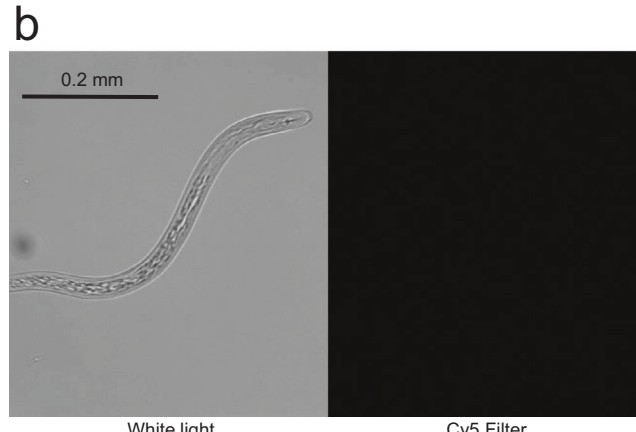

**Fig. 8 The ability of Cry14Ab to reduce *Heterodera glycines* infection in roots of plants grown in the greenhouse was assessed. a** Western blot of 10 individual T2 homozygous plants from event SB166, showing consistent accumulation of Cry14Ab protein. A second blot of another 9 plants showed similar results. **b** *H. glycines*-infested roots of soybean negative control event SB172, stained pink with acid fuchsin 10 days post infestation. Four plants were imaged, of which one is shown. **c** *H. glycines*-infested roots of soybean Cry14Ab event SB166, stained pink with acid fuchsin 10 days post infestation. Four plants were imaged, of which one is shown. Source data are provided as a Source Data file.

**Fig. 7 Microscopic observation of ingestion of fluorescently labeled Cry14Ab by *Heterodera glycines*. a** Two second-stage juvenile *H. glycines* nematodes were fed IR680-labeled Cry14Ab at 1 mg/mL in the presence of the feeding stimulant octopamine for 24 h. Each nematode is shown under white light on the left, and with a Cy5 filter set on the right. The experiment was performed twice with 200 nematodes under each condition. About half ingested labeled protein, and three representative nematodes were photographed each time, of which two are shown here. **b** A nematode that was not fed labeled Cry14Ab, under white light on the left, and with a Cy5 filter set on the right. Source data are provided as a Source Data file.

Cry14Ab soybean events described in this study were generated by co-transforming plants with two separate vectors that contained the *cry14Ab1* gene on one vector, and the selectable marker herbicide resistance gene *grg23ace5* on the other (Supplementary Fig. 2). The presence of the *cry14Ab* gene in regenerated transformed plants was confirmed by PCR using primers Cry14Ab_p2f and Cry14Ab_p3r (see Supplementary Table 1),

which amplify a fragment of the gene, and its expression was confirmed by western blot. Plants that contained the gene and had a detectable expression of the protein were allowed to self, and the seed was used in greenhouse and field *H. glycines* efficacy testing.

T2 homozygous plants showed consistent levels of accumulation of the Cry14Ab protein (Fig. 8a). *H. glycines* developing on roots expressing Cry14Ab or on negative control roots were stained using a standard acid fuchsin protocol[33] 10 days post infestation (Fig. 8b and c). More nematodes were found developing on the roots of the negative control plants than on the roots of the plants expressing Cry14Ab, suggesting that Cry14Ab interferes with the ability of *H. glycines* to establish and maintain infection sites. In addition, it was found that *H. glycines* development on plants expressing Cry14Ab was significantly delayed compared to development on control plants. Most nematodes on negative control roots appeared to be at the J3 or J4 stage 10 days post infestation, while most nematodes on the plants expressing Cry14Ab appeared to be at the J2 or J3 stage. This suggests that in addition to inhibiting the establishment of infection, Cry14Ab retards or halts the development of juvenile *H. glycines*.

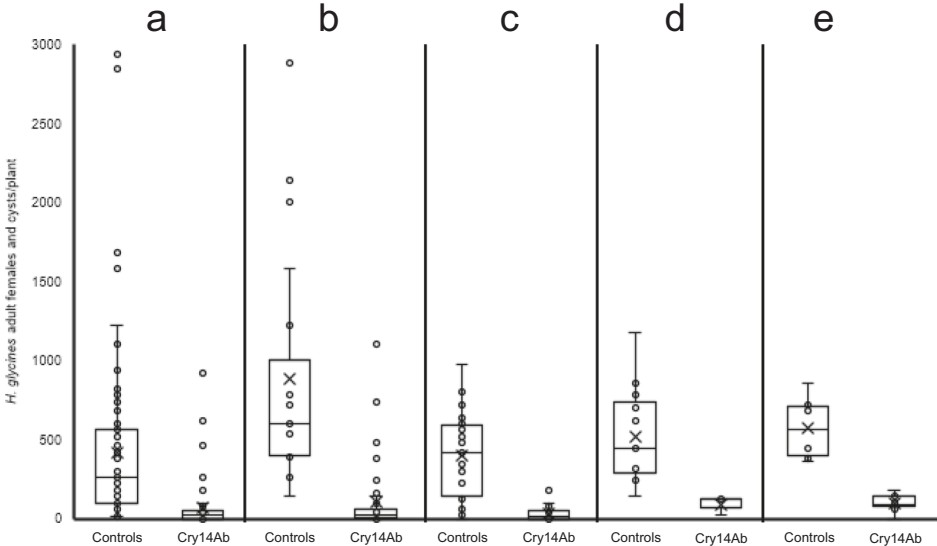

**Fig. 9 The ability of Cry14Ab to reduce *Heterodera glycines* reproduction in the greenhouse was assessed in the PI 88788-derived soybean variety Jack.** Untransformed plants and negative control events transformed only with a selection vector (Controls) were compared to multiple Cry14Ab-expressing events (Cry14Ab). Soybean plants were grown in sand in 4-inch clay pots in a greenhouse and infested at 2 weeks with *H. glycines*. Box plots indicate median (middle line), average (*x*), 25th to 75th percentile (box) and 5th and 95th percentile (whiskers) as well as all data points. **a** T1 soybean plants (Controls: $n = 82$ plants; Cry14Ab: $n = 57$ plants) were infested with *H. glycines* line OP50, and after 30 days adult females and cysts were counted, giving FI = 14.7 ($P < 0.0001$, $t = 4.8$, df = 137). **b** T1 soybean plants (Controls: $n = 19$ plants; Cry14Ab: $n = 36$ plants) were infested with *H. glycines* line OP50, and after 60 days adult females and cysts were counted, giving FI = 12.3 ($P < 0.0001$, $t = 5.1$, df = 56). **c** T2 soybean plants (Controls $n = 41$ plants; Cry14Ab: $n = 47$ plants) were infested with *H. glycines* line OP50, and after 60 days adult females and cysts were counted, giving FI = 8.4 ($P < 0.0001$, $t = 10.0$, df = 82). **d** T1 soybean plants (Controls: $n = 15$ plants Cry14Ab: $n = 3$ plants) were infested with HG type 2.5.7 *H. glycines*, and after 30 days adult females and cysts were counted, giving FI = 16.8 ($P = 0.03$, $t = 2.3$, df = 16). **e** T2 soybean plants (Controls; $n = 6$ plants; Cry14Ab: $n = 8$ plants) were infested with HG type 2.5.7 *H. glycines*, and after 60 days adult females and cysts were counted, giving FI = 17.0 ($P < 0.0001$, $t = 6.2$, df = 12). Source data are provided as a Source Data file.

Standard greenhouse methods have been developed to evaluate new soybean varieties for *H. glycines* resistance[34]. In a standard assay, the degree of reproduction of *H. glycines*, known as the female index (FI), is measured by the ability of female nematodes to grow and reproduce on a standard *H. glycines*-susceptible soybean check cultivar (typically Lee 74, Essex, Hutcheson, or Williams 82) and on the test cultivar, grown and infested in parallel. The FI value is calculated from at least three replicated results using the formula: (mean number of females on test cultivar)/(mean number of females on check cultivar)×100. By convention cultivars with a FI < 10 are considered resistant, and those with a FI between 10 and 30 are considered moderately resistant. We utilized the transformation background Jack as the check cultivar in our greenhouse assays. Jack contains native resistance to *H. glycines* derived from PI 88788, and using it as the check cultivar allowed us to measure the efficacy of Cry14Ab in the presence of native resistance. Using the standard methods[34], we assessed the efficacy of Cry14Ab against two *H. glycines* populations, both of which are considered virulent to PI 88788-derived resistance. The first population was an inbred line OP50[35] (HG type[36] 1.2.3.5.6), and the second was a field population of HG type 2.5.7 obtained from the University of Illinois.

*H. glycines* reproduction after 30 or 60 days (one or two *H. glycines* generations) was compared on Jack expressing Cry14Ab to reproduction on either (1) untransformed Jack, (2) F2 negative segregants, or (3) negative control events transformed only with a selection vector (pAX5219). The results are shown in Fig. 9. The degree of *H. glycines* resistance contributed by Cry14Ab was expressed as FI = (mean number of females on Cry14Ab event)/(mean number of females on untransformed Jack, negative segregants, or negative control events) × 100. Cry14Ab events with FI < 10 were categorized as highly resistant to *H. glycines*,

while events with FI of 10–30 were categorized as moderately resistant to *H. glycines*[34]. Note that many commercially available soybean varieties labeled as *H. glycines* resistant do not meet the FI < 10 standard in greenhouse testing, so while this value can indicate *H. glycines* resistance, results in the field appear to depend on many additional factors.

In this study, seven independent Jack events that had a consistent expression of Cry14Ab were produced and were tested as described above to determine if Cry14Ab was efficacious against *H. glycines*. Events that accumulated Cry14Ab protein based on western blot analysis had significantly fewer *H. glycines* females on their roots than plants that did not contain the Cry14Ab gene (either Cry14Ab-negative segregants, untransformed Jack, or negative control events SB171 or SB172 that had been transformed with pAX5219 only). Cry14Ab reduced infection by both the OP50 and HG type 2.5.7 *H. glycines* populations. To test whether the Cry14Ab trait was stable in the T2 soybean generation, the *H. glycines* greenhouse assays were repeated using plants grown from segregating T2 hemizygous seed from some events. The results revealed that the gene was transferred to the next generation, and continued to provide resistance to *H. glycines*.

**Field evaluation of efficacy of Cry14Ab against *H. glycines* in soybean.** Field trials were conducted in Webster County, Iowa in 2010 and 2011. We used standard practices for field evaluations of *H. glycines*-resistant cultivars, which call for a minimum of three replications and two years of data[37]. The site was chosen based on the presence of a *H. glycines* population virulent to PI 88788-derived resistance in the Jack soybean variety. Five Cry14Ab events were planted along with two control treatments.

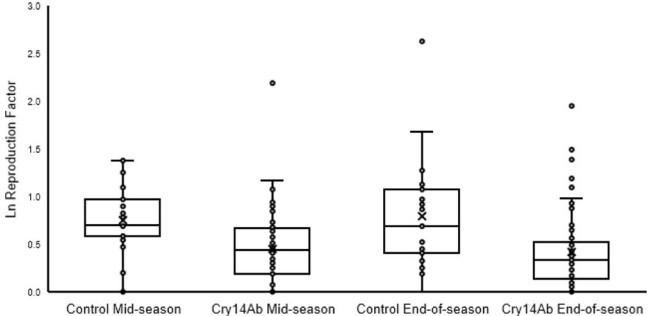

**Fig. 10 The ability of Cry14Ab to reduce _Heterodera glycines_ reproduction in the field was assessed in the PI 88788-derived soybean variety Jack.** Untransformed plants and negative control events transformed only with a selection vector (Control) were compared to multiple Cry14Ab-expressing events (Cry14Ab). Soybean plants were grown in a _H. glycines_-infested field in Webster County, IA in 2010 and 2011. Soil core samples were taken to assess the population density of _H. glycines_ at planting, ~60 days after planting (mid-season), and at harvest (end-of-season). A reproduction factor was calculated for mid-season ($n = 26$ plots for control, $n = 66$ plots for Cry14Ab) and end-of-season ($n = 26$ plots for control, $n = 65$ plots for Cry14Ab), by dividing the population density at the assessment time by the population density at planting. The natural log of the reproductive factor plus one is displayed. Data were natural log (ln) transformed and analyzed with an ANOVA. Contrast statements were used to compare control and Cry14Ab treatments with an _F_-test. Cry14Ab-expressing soybean reduced _H. glycines_ reproduction at both the mid-season ($P = 0.00179$) and end-of-season ($P = 0.00156$) assessment times. In examining non-transformed data, mid-season reproduction was reduced by 43%, from a reproduction factor of $1.25 \pm 0.8$–$0.71 \pm 1.06$ and end-of-season reproduction was reduced by 60%, from a reproduction factor of $1.67 \pm 2.52$–$0.67 \pm 1.0$. Box plots depict the median (middle line), average (_x_), 25th and 75th percentile (box), and 5th and 95th percentile (whiskers) as well as all data points. Source data are provided as a Source Data file.

The control treatments consisted of untransformed Jack, and Jack transformed with the herbicide tolerance gene (_grg23ace5_) alone. The experimental treatments were planted in three-row plots arranged in a randomized complete block with four replications. Soil samples were collected from each experimental plot to determine the initial, mid-season, and end-of-season _H. glycines_ egg population densities in all plots. A reproductive factor was calculated to evaluate nematode reproduction mid-season and end-of-season. Reproduction factors were calculated by dividing the mid-season and end-of-season population densities by the initial population density, respectively. A summary of these results is shown in Fig. 10. The results are expressed as the average reproduction factor for all Cry14Ab events compared with the reproduction factor for the Jack controls. Reproduction of _H. glycines_ was significantly lower in Cry14Ab-containing events compared to control treatments at both mid-season ($df = 1, 77$; $F = 10.447$; $P = 0.00179$) and end-of-season ($df = 1, 71$; $F = 10.742$; $P = 0.00156$), illustrating the potential for Cry14Ab to be used to control _H. glycines_ in commercial soybean crops.

## Discussion

No transgenic agricultural products are available for plant-parasitic nematode control. Previously Cry6A (54 kDa) and a truncated form of Cry5B (79 kDa) have been shown to control root-knot nematode (_Meloidogyne incognita_) in tomato hairy roots in laboratory experiments[15,38]. Here we show that Cry14Ab controls _H. glycines_ in the field when expressed in soybean plants.

Cry14Ab was identified by using _C. elegans_ as a model for _H. glycines_. PPNs differ from _C. elegans_ in that, while _C. elegans_ can

ingest and grind entire bacteria, PPNs can only ingest through a narrow stylet that limits the size of the material that can pass through[39]. Furthermore, _H. glycines_ produces a feeding tube[40] within the large multinucleate plant root cell that the nematode induces the plant to form and that serves as its feeding site[41], and all nutrients that will be ingested by the nematode must pass through that feeding tube. Previous work on the injection of fluorescent proteins into the multinucleate plant cell feeding sites seemed to indicate that large proteins may not be ingested by _H. glycines_ or other PPNs[42]. However, published results suggest that the ability of a protein to be ingested by _H. glycines_ may depend on more than just its molecular weight, and may be influenced by a combination of the protein's size, shape, and electrostatic charge[43].

In this study, we have shown that a purified protein with a molecular weight of 132 kDa can be ingested by J2 _H. glycines_ in vitro. Fluorescently labeled Cry14Ab was found in the esophagus and intestine of J2 _H. glycines_ nematodes after 24 h of incubation in the presence of the feeding stimulant octopamine in a laboratory experiment done in liquid. J2 _M. hapla_ nematodes have also been shown to be able to ingest purified proteins as large as 140 kDa in laboratory experiments done in liquid[44]. When the full-length Cry14Ab protein is expressed in transgenic soybean plants the number of _H. glycines_ that reach maturity on the plants is reduced, although it remains to be determined if the nematodes are ingesting the full-length protein in roots. It also remains to be determined if the nematodes ingest Cry14Ab after establishing plant cell feeding sites in the roots, and/or while they are moving through the roots, searching for locations to induce the formation of feeding sites.

In _C. elegans_ this protein exerts its toxic effects on the intestine, which is also true of Cry5B[19]. Insecticidal Cry toxins also damage insect intestinal cells by inserting into the apical membrane to form pores or ion channels[26]. While we do not have direct evidence that Cry14Ab affects the intestine of _H. glycines_, that seems likely to be the case because of the many ways the protein is similar to other Cry toxins.

Most Cry toxins are easily cleaved by trypsin or other proteases to an active protease-resistant core with a molecular weight of about 55–70 kDa[27]. By contrast, Cry14Ab is not cleaved to a protease-resistant core in vitro by the proteases that were tested in this study at pH values between 5 and 10.4. The control of _H. glycines_ described here was achieved by expressing full-length protein in soybean plants, although it remains possible that the nematodes create conditions that lead to proteolysis either in the plant cell feeding site or in the nematode itself. Overall Cry14Ab appears similar to insecticidal Cry toxins in terms of its amino acid sequence, its ability to form crystals, and its ability to associate with and damage intestinal cells (in this case _C. elegans_ intestinal cells).

The mechanism of action of Cry14Ab is likely to be fundamentally different from the putative mechanism of the native trait found in PI 88788, which is believed to work by interfering with vesicular trafficking in feeding sites, preventing the nematodes from continuing to feed[45]. By contrast Cry14Ab most likely directly damages the intestines of the nematodes. This can explain why Cry14Ab controls _H. glycines_ HG types that are virulent to PI 88788. Combining Cry14Ab with PI 88788 in soybean plants should delay the development of virulence in _H. glycines_ to both of these control mechanisms.

Soybean farmers are in need of new methods to control _H. glycines_, one of the principal causes of damage in the crop. Expression of Cry14Ab in soybean plants is a potential approach to controlling this pest, and as such may help farmers deal with the growing virulence of _H. glycines_ to existing control methods.

## Methods

**Bacterial strain isolation from environmental samples.** Environmental samples of soil were collected from many locations. Soil samples were plated onto the surface of Luria Bertani (LB) agar and individual colonies were selected and transferred onto fresh LB agar. Additional re-streaks were performed until visually uniform, purified strains were obtained. Liquid cultures of purified strains were stored at −80 °C, and these frozen stocks were used to screen for strains that showed activity in *C. elegans* bioassays.

**C. elegans bioassays.** Standard methods were used to maintain N2 *C. elegans* cultures as described in WormBook[22]. Nematodes were maintained on MYOB plates that had been inoculated with *E. coli* strain OP50. Nematodes were washed off plates, rinsed, floated on 20% sucrose with 0.1 M NaCl, rinsed again, and resuspended in S media.

Bioassays were performed by adding 10 μL of the sample to be tested to a well of a 96-well plate, along with 0.5 μL *E. coli* worm food (strain OP50 or HB101), 0.5 μL tetracycline at 5 mg/mL in 95% ethanol, 0.15 μL chloramphenicol at 20 mg/mL, 83.85 μL S media, and 5 μL *C. elegans* in S media containing ~50 worms of mixed life stages. Negative control samples consisted 10 μL of sterile microbial growth medium when bacterial strains were being assayed, or 10 μL of the appropriate buffer when purified proteins samples were being assayed. Plates were covered and put on a shaker for 3–5 days at 225 rpm at room temperature. The bioassays were scored visually using a dissecting microscope as shown in Fig. 1, with scores ranging from 0 to 5. A score of 0 indicated that the well resembled negative control wells, with hundreds of very active nematodes and almost complete clearing of the *E. coli* that had been added as worm food. A score of 5 indicated that only a few, sluggish worms were present, and the well was cloudy with unconsumed worm food.

**Purification of Cry14Ab from native strain.** The native strain was grown in 100 mL of T3 medium at 30 °C for 5 days with shaking at 250 rpm. The culture was centrifuged to pellet the cells, and the supernatant was passed through a 0.2 μm filter, and dialyzed against 20 mM Tris pH 8, 20 mM NaCl. The dialyzed supernatant was fractionated on a Mono Q anion exchange column using a 0–1 M NaCl gradient. Fractions were dialyzed against 25 mM Tris pH 7, 25 mM NaCl before being bioassayed against *C. elegans*. An active fraction was concentrated 5 fold in a 3 kDa molecular weight cutoff Centricon concentrator, run on SDS–PAGE, and blotted to PVDF using standard methods. Bands were cut out and N-terminal sequencing was done by the Protein Facility of Iowa State University.

**Heterologous expression of Cry14Ab.** For *E. coli* expression, the gene was cloned into the pMAL vector system (New England Biolabs), resulting in a protein with a 6-histidine and maltose-binding protein tag at its N-terminus. The vector was then used to transform the *E. coli* cell line One Shot BL21 Star (DE3) (Invitrogen #C6010-03). A single colony was picked and grown in LB media plus carbenicillin with shaking at 37 °C. Once cells reached an $OD_{600}$ of 0.8, IPTG was added to a concentration of 50 μM and the temperature was reduced to 20 °C. The culture was allowed to grow overnight and was then harvested by centrifugation. Cell pellets were frozen for storage prior to lysis. Cell pellets were thawed and resuspended in 20 mM Tris pH 8, 200 mM NaCl, 1 mM EDTA (Buffer A) at 1/10th the original culture volume. Lysonase (Novagen 71-230-4) was added at a concentration of 2 μL per milliliter of resuspended volume, and the cells were left to lyse at room temperature for 45 min. The cells were then refrozen and thawed just prior to purification.

Lysed cultures were clarified by centrifugation to pellet cell debris and then filtered through a PES 0.2 micron filter. Lysates were then purified using an AKTA FPLC system over an MBPTrap HP column (GE Healthcare, 28918779). Sample was loaded onto the column through a sample loop at a flow rate of 0.5 mL/min. Bound proteins were washed with 5 column volumes of buffer A, and Cry14Ab was eluted with 5 column volumes of Buffer A with 10 mM maltose. Fractions containing protein were pooled and dialyzed against 20 mM Tris pH 8.0 overnight with 2 changes of buffer, 4 L each.

The protein was quantified by running SDS–PAGE gels using standard methods, with known quantities of BSA standard protein (Pierce #23209) in adjacent wells. Gels were stained with Invitrogen SimplyBlue SafeStain (Thermo Fisher #LC6065). Biorad Image Lab software, version 6.0.0, was used to estimate the concentration of Cry14Ab compared to the BSA standards.

For *Bt* expression, Cry14Ab was cloned into a *Bt* expression vector with a Cry1Ca promoter and a Cry1Ac terminator[24], which was passed through a $dam^−$/$dcm^−$ *E. coli* strain before being used to transform *Bt* Cultures were grown in CYS media with 10 μg/mL tetracycline. Sporulated cultures were pelleted, and Cry14Ab protein was extracted by sonication and shaking in 20 mM CAPS pH 10.5, 10 mM beta-mercaptoethanol, 10% glycerol. For bioassays the protein was dialyzed to remove beta-mercaptoethanol and glycerol. For further purification, the protein was fractionated on a Superdex 200 column in the extraction buffer. The protein concentration was determined by running Coomassie-stained SDS–PAGE with known quantities of BSA for comparison.

To raise polyclonal antibodies against Cry14Ab, a version of the protein with a 6-his tag at the N-terminus was expressed in *E. coli*, and was extracted from a cell pellet in 50 mM Tris pH 8. After dialysis into 50 mM sodium phosphate pH 7, 300 mM NaCl, 10 mM imidazole, the protein was purified using cobalt resin. The purified protein was concentrated to about 4 mg/mL and was run on SDS–PAGE with a large enough well to hold 1.7 mg of the protein. After Coomassie staining, the band corresponding to Cry14Ab was cut out and sent to Pacific Immunology for injection into rabbits.

To make truncated Cry14Ab, the glycine at position 708 was mutated to arginine. The mutant protein was expressed in *Bt* as for wild-type protein and was cleaved with trypsin. The N-terminal fragment of the cleaved protein was purified by anion exchange chromatography and was of the expected size based on SDS–PAGE, and an aliquot was fluorescently labeled as for wild-type protein.

**Proteolysis of Cry14Ab.** Cry14Ab was expressed in a plasmid-cured strain of *Bt* grown in liquid medium to sporulation and then pelleted. The spore/crystal pellets were extracted with 20 mM CAPS pH 10.5, 10 mM beta-mercaptoethanol, 10% glycerol, and the supernatant was run on a Superdex 200 gel filtration column in the same buffer. Fractions containing Cry14Ab were pooled and concentrated using Centricon centrifugal filter units with a molecular weight cutoff of 10 kDa. Purified protein at 0.4 mg/mL was incubated with 0.004 mg/mL of each protease (1:100 wt:wt ratio of protease to Cry14Ab) using Hampton Research Proti-Ace and Proti-Ace 2 kits, in 20 mM sodium acetate pH 5, 0.15 M NaCl, or 5 mM HEPES pH 7.5, 0.25 M NaCl, or 20 mM CAPS pH 10.4, 0.15 M NaCl, for 2 h at 37 °C. The reactions were halted by heating to 95 °C for 10 min in SDS–PAGE sample buffer with β-mercaptoethanol. 10 μL of each sample was run on SDS–PAGE and the gels were stained with Coomassie blue.

**Visualizing the effect of Cry14Ab on the nematode intestine.** BSA was labeled with Alexa Fluor 680 NHS Ester (Invitrogen). Cry14Ab was labeled with IRDye 680RD NHS Ester (LI-COR Biosciences). To test whether the label had affected the activity of Cry14Ab, labeled and unlabeled protein were tested in bioassays in a 2-fold serial dilution with protein concentrations ranging from 1 to 0.008 mg/mL. About 600 *C. elegans* in 400 μL of S media containing 50 μg rifampin plus HB101 *E. coli* worm food were mixed with 100 μL of labeled protein to give a final protein concentration of 0.066 mg/mL. To detect ingestion the nematodes were incubated for 4 h, pelleted and rinsed with S media three times, anesthetized with 1% 1-Phenoxy-2-propanol, and imaged with white light or with a Cy5 filter set. To detect retention of Cry14Ab in the nematode intestine, the nematodes were incubated for 24 h, pelleted and rinsed with S media three times, incubated in S media for another 24 h, and imaged as before.

**Production of transgenic soybean expressing Cry14Ab.** For the expression of Cry14Ab in soybean two separate vectors were constructed for co-transformation, while a third vector used alone to generate negative control plants (Supplementary Fig. 2). The first vector, pAX5347, contained the gene for Cry14Ab and the gene for a yellow fluorescent protein (YFP) marker. The second vector, pAX5207, contained the gene for the glyphosate selectable marker GRG23Ace5 (a modified 5-enolpyruvylshikimate-3-phosphate synthase), and the gene for phosphomannose isomerase (PMI). The gene for Cry14Ab was re-coded from the native sequence to a soybean-optimized version. These two vectors were used to co-transform soybean cv. Jack using aerosol beaming of embryogenic callus[46,47], and plants were selected using the glyphosate resistance gene as a selectable marker during event regeneration. Negative control plants were generated by transforming soybean with the third vector, pAX5219, which contained the same gene for a glyphosate selectable marker (GRG23Ace5) and the gene for green fluorescent protein 2 (GFP2), and plants were again selected using the glyphosate resistance gene as a selectable marker during event regeneration.

**Cry14Ab detection by western blot.** Leaf samples (0.5 cm leaf disks) were taken from full, newly emerged leaves and maintained at −80 °C prior to processing. Samples were processed in 1.5 mL Eppendorf tubes by adding 200 μL of NuPAGE LDS sample buffer (Invitrogen NP007) with 2.5% beta-mercaptoethanol along with small stainless steel beads, and disrupted for 1 min in a Mini-Beadbeater-96 (Bio Spec Products), followed by centrifugation. Soluble material was transferred to a clean tube and heated to 70 °C for 10 min. SDS–PAGE was run and proteins were transferred to a nitrocellulose membrane. The molecular weight standard was PageRuler™ Unstained Protein Ladder (Thermo Scientific catalog number 26614). Rabbit antibodies raised against Cry14Ab (described above) were used as the primary antibody with a 1:500 dilution. The secondary antibody was HRP-Conjugated Donkey Anti-Rabbit IgG (H+L) Cross-Adsorbed Secondary Antibody (Invitrogen Catalog # SA1-200) diluted 1:10,000. A chemiluminescent detection reagent was used (Pierce ECL Western Blotting Substrate #32106). Blots were exposed to film in a radiography cassette for 10 min.

**H. glycines greenhouse testing.** The *H. glycines* greenhouse assay used in this study consisted of T1 or T2 soybean seeds germinated in 4-inch clay pots containing sand plus Osmocote or Multicoat 4. All plants were grown for 3–4 weeks prior to being infested with ~10,000 *H. glycines* (line OP50, or HG type 2.5.7 field strain) at the J2 stage, and allowed to progress through either one or two *H. glycines* lifecycles (30 or 60 days, respectively). At the end of the assay, plants were removed

from the sand and the females were harvested and counted for each individual plant. Statistical analysis was done using R, version 3.5.2. Graphs were produced using Microsoft Excel, version 2008.

**Field trials**. A field plot experiment was planted in Webster County, IA in 2010 and 2011. Control and experimental event plants were planted in plots consisting of three rows 0.3 m long and spaced 76.2 cm apart. Plots were spatially isolated from each other with 1.8 m borders to prevent soybean roots from growing between plots. Plots were arranged in a randomized complete block design with four replications and were grown using normal cultural practices. For each entry, four replications of 24 plants each were planted. After PCR analysis, non-transgenic segregates were removed by hand. Controls in the trial were untransformed Jack, Jack transformed with the herbicide tolerance gene (grg23ace5) alone (also present in test plants), and a variety with no known resistance to any HG type of H. glycines (Lee74 or Thorne). Soil samples collected from the middle of each plot were taken to estimate the population density of H. glycines. A soil sample consisted of six soil cores measuring 1.9 cm in diameter and taken to a depth of 15 cm. H. glycines eggs were extracted and counted from a representative 100 cc subsample of each six-core soil sample to determine the population density of H. glycines for each sample[9]. One control plot and two Cry14Ab plots were lost after planting, and one Cry14Ab plot was lost after mid-season soil sampling. Statistical analysis was done using R, version 3.5.2. Graphs were produced using Microsoft Excel, version 2008.

**Reporting summary**. Further information on research design is available in the Nature Research Reporting Summary linked to this article.

## Data availability

Data supporting the findings of this work are available within the paper and its Supplementary Information files. A reporting summary for this article is available as a Supplementary Information file. Cry14Ab is GenBank accession AGU13817.1 [https://www.ncbi.nlm.nih.gov/protein/533206134]. RefSeq, the NCBI Reference Sequence Database, is available at https://www.ncbi.nlm.nih.gov/refseq/. The datasets and materials generated and analyzed during the current study are available from the corresponding author upon request, and a response can be expected within 2 months. The proteins, associated genes, and other aspects of the concept described in this article are protected under one or more patent filings and may also be subject to other intellectual property rights. BASF reserves the right to require a requester of such materials to enter into a non-disclosure agreement, a material transfer agreement, or other common type of agreement, in order to receive the materials. Further, some of the materials used to generate the data in this paper are highly regulated by government agencies and can only be shared with parties that meet all of BASF's regulatory and stewardship requirements. The use of the materials will be limited to non-commercial research uses only. Source data are available in figshare [https://doi.org/10.6084/m9.figshare.14515587.v1][48]. Source data are provided with this paper.

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

## Acknowledgements

We thank Jun Cao for performing the plant transformations, Jayme Williams for preparing the truncated version of Cry14Ab, Jeanett Perez-Lesher for preparing vector illustrations, and Cheryl Peters for technical support.

## Author contributions

T.W.K., N.B.D., and J.D. contributed to the conception, design, and interpretation of the study; T.W.K., L.C.S., K.S., J.Z., and J.D. performed the experiments; T.W.K., M.T.M., L.C.S., K.S., J.Z., and J.D. contributed sections to the manuscript; T.W.K. and J.D. wrote and compiled the final manuscript. All the authors reviewed the manuscript.

## Competing interests

At the time this work was carried out the authors were employees of BASF, a for-profit company. BASF holds or has applied for patents and/or other intellectual property rights on the proteins, associated genes, and other aspects of the concept described in this article.
