## [Peer Review File · Nature Communications]

REVIEWER COMMENTS

Reviewer #1 (Remarks to the Author):

This manuscript describes what is now the relatively straightforward procedure of cloning a *Bacillus thuringiensis* toxin gene and expressing it in a plant to provide resistance against a given pest. From the title onwards the authors are claiming that there are many novel aspects to this particular work which I feel are rather oversold. For instance, Cry proteins (including Cry14) have long been known to have nematocidal activity, including activity to root knot nematodes. Furthermore, such proteins have also previously been expressed in planta to provide protection. The novelty only really stems from the fact that Cry14Ab rather than Cry14Aa is being used and the expressing plant is soybean rather than tomato.

The science itself is fine and appears to confirm that Cry14Ab is acting in the way one would expect from a Bt Cry toxin. The manuscript though does contain some additional claims and inferences which require closer scrutiny:

1) Lines 97-99. The antibiotic experiment described here is potentially opening a can of worms (if you will excuse the pun). There has been a significant amount of literature on the effects of antibiotics on Bt toxin efficacy concentrating on the use of antibiotics to remove native gut flora from the target pest. The observation that antibiotics don't affect the activity of the strain could in fact arise from competing effects of eliminating gut flora and inhibiting Bt growth. Better controlled experiments are required to support the stated conclusion. The obvious one would be to test purified crystals.

2) The manuscript often makes the claim that unlike other Cry toxins, Cry14Aa is soluble at pH8. It is not at all clear what the authors mean by this. If you were to take a Bt toxin crystal (eg from the Cry1Ac expressing strain HD73), solubilize it at pH10.5 (say) and then dialyse in a buffer at pH8 the protein will remain soluble. This is true of most Cry proteins. What is certainly not true is that these Bt crystals will solubilize at pH 8, however I can see no evidence that this is true for the Cry14Aa crystals either. All solubilization experiments in this manuscript appear to have been performed at pH10.5

3) The above claim might therefore just refer to the fact that some Cry14Aa was found in a soluble form in the sporulated/sporulating Bt culture which was grown at pH8. Although the authors state that this is not the case for other Cry toxins they do not provide evidence or citations to support this. The toxin will probably be produced in a soluble form within the bacterium before being laid down as a crystal and it is quite conceivable to imagine that some of the toxin will not be incorporated into the crystal. To claim that this is specific to Cry14Ab the authors should provide more evidence that such a situation is not seen with other Bt strains. Possibly related to this is the observation that 30ml cultures had activity whereas 1-2 ml ones didn't. This needs further explanation / experimentation.

4) From the description of the methodology I presume that the authors were screening supernatants of Bt strains looking for novel virulence factors and came across the toxin through this route. Presumably Cry14Ab is also found in the crystal of the strain being characterized? This should be made clear.

5) The data provided in the manuscript certainly indicate that Cry14Ab is not proteolytically processed at the C-terminus by the range of proteases used. As the authors point out the conditions in the nematode gut may provide an environment where cleavage can happen. Having presented all the protease resistance data the reader will need to be told what happens to the protein (proteolytically speaking) within the gut. I suspect that the authors attempted such an experiment but without getting clear data and would suggest that either clear data are obtained, or this section removed.

Aside from the above I also have a number of minor points for consideration

6) Line 65. When referring to toxins that “only affect nematodes that feed on those plants” the authors should clarify the point that I believe that they are making - that the protein may affect other nematodes but will not have an effect in the field since those nematodes will not be exposed to the toxin.

7) Fig1. Box d is meant to show sporulated Bt yet I can't see any spores.

8) Fig3. I would prefer that the X-axes of boxes c and d have an identical range

9) Fig3. The graph itself (rather than just the legend) should state the source of the protein

10) Line 270 spelling of trypsin

11) Line 463 you can't say that it doesn't have activity against other tested insects without naming those insects.

12) 12 Line 512 spelling of grown

13) References – lots of italicization missing

14) Ref 16 name?

Reviewer #2 (Remarks to the Author):

This manuscript by Kahn and colleagues describes the isolation of a novel endotoxin, Cry14Ab, from the bacterium *Bacillus thuringiensis*. The authors used various bioassays to demonstrate that Cry14Ab is highly active against free-living nematode *C. elegans*. Furthermore, plants that expressed Cry14Ab showed significant levels of protection from plant-parasitic cyst nematodes in both greenhouse and field trials. Overall, I believe that this work is of significant importance, and I see it as a potentially good fit for *Nature Communications*. However, a number of technical and conceptual issues need addressing before this manuscript will be suitable for publication.

Main Remarks:

(1) Previous size exclusion experiments have shown that the feeding tube produced by cyst nematodes can only take up molecules approx. 30 kDa and smaller (Goverse et al., 1998; Bockenhoff et al., 1994; Urwin et al., 1997), but this paper states that a 132 kDa (Cry14Ab) protein is able to enter into the nematode gut. A possible reason could be that mechanical damage caused by nematodes inside roots activates a novel set of proteases that cleave Cry14Ab to a protease-resistant core, which is in turn ingested by nematodes. Considering environmental concerns related to use of Bt, such a possibility should be tested and discussed in the manuscript.

(2) Fig. 3: Important experimental details have been omitted, making it difficult to judge the rigor and reliability of the data. What do the error bars indicate? Why do some data points have them but others not? How many technical and biological replicates were performed? What is the statistical significance of the data? Please provide all such information in the figure legend. Also, note that bioassay is misspelled and that the graphs are of suboptimal quality, with the y-axis less a legend than a title.

(3) Fig. 5: What is the statistical relevance of the data? What do the error bars represent? Why do only some data points have error bars? Why is there a difference between truncated unlabeled and truncated labeled? At concentration 0 there seems to be two samples for truncated unlabeled (diamonds). I suggest using a colored graph to better define the data points. Also, why were all four proteins not tested at the various concentrations?

(4) Fig. 9: Consider splitting the graphs into F1-30 day, F1-60 day, and F2-60 day or perhaps just simplifying by using only one time point. Second time point can go into supplementary data. How

many plants were used (n =?)? Again, explain the error bars.

(5) Fig. 10: Crucial details are missing in the figure legends. Also, how many plants were tested (n =?).

(6) Even though the materials and methods section discusses methods of identifying and isolating the bacterial strain, more detail is needed on how to do so (lines 499–506). How was the strain actually isolated? Data from worm bioassays should be added to this section to support these findings.

(7) Many abbreviations were used throughout the manuscript without clearly establishing their meanings. Readers are likely to be confused at being left to search for abbreviations' meanings themselves (cf. line 135).

(8) Lines 615–638, the field trial description, currently included in the materials and methods section, should perhaps also be included in the results section.

(9) The word bioassays is used multiple times without citation and without explanation of how bioassays were performed (lines 101–116).

(10) A number of key observations and conclusions are arrived at abruptly without proper citations (cf. lines 144–145, 150–151). A thorough revision of literature is recommended.

Minor Remarks:

(1) Fig. 1: Please provide scale bars for all microscopic images. Also, it is unclear exactly what these figures show? What do a, b, c and d indicate? Please label crystals in each image.

(2) Fig. 4: Using different arrangements of proteases and ladder in three different gels may confuse readers.

(3) Figs. 6 and 7: Please add scale bars. Also, I suggest placing figure 7 earlier in the manuscript.

Shahid Siddique
UC Davis

We thank the reviewers for their comments and suggestions, which helped us make significant improvements to the manuscript. Here are our responses and the changes we have made to the manuscript. The reviewers' comments are in black, and our responses are in blue.

Reviewer 1.

General comment.

This manuscript describes what is now the relatively straightforward procedure of cloning a *Bacillus thuringiensis* toxin gene and expressing it in a plant to provide resistance against a given pest. From the title onwards the authors are claiming that there are many novel aspects to this particular work which I feel are rather oversold. For instance, Cry proteins (including Cry14) have long been known to have nematocidal activity, including activity to root knot nematodes. Furthermore, such proteins have also previously been expressed in planta to provide protection. The novelty only really stems from the fact that Cry14Ab rather than Cry14Aa is being used and the expressing plant is soybean rather than tomato.

Previously Cry5B has been shown to control root knot nematode in transgenic tomato hairy roots. Hairy roots are an artificial system that can only be studied in a laboratory setting. The novelty of our studies stems from the fact that this is the first time a three-domain delta-endotoxin protein, or any protein, has been shown to control nematodes in transgenic whole plants. We have demonstrated control in plants grown in the greenhouse, and more importantly, in the field. Furthermore, the nematode that is controlled by Cry14Ab is soybean cyst nematode (*H. glycines*), a major agricultural pest in one of the most important and widely grown crops in the world. There is currently no transgenic approach to controlling this nematode. Soybean is a crop in which it is practical to use a transgenic approach for pest control (unlike tomatoes, which would require many varieties to be transformed and approved, rendering this approach uneconomical). Because soybean is one of the major sources of food and feed in the world, new methods to control one of its most damaging pests are badly needed.

1) Lines 97-99. The antibiotic experiment described here is potentially opening a can of worms (if you will excuse the pun). There has been a significant amount of literature on the effects of antibiotics on Bt toxin efficacy concentrating on the use of antibiotics to remove native gut flora from the target pest. The observation that antibiotics don't affect the activity of the strain could in fact arise from competing effects of eliminating gut flora and inhibiting Bt growth. Better controlled experiments are required to support the stated conclusion. The obvious one would be to test purified crystals.

We have changed lines 97-99 to explain that while we can't say for sure that the results of the antibiotics experiment arose from inhibition of the growth of the strain, the result was enough to encourage us to continue trying to isolate an active protein from the strain. This was the only purpose of the experiment.

2) The manuscript often makes the claim that unlike other Cry toxins, Cry14Aa is soluble at pH8. It is not at all clear what the authors mean by this. If you were to take a Bt toxin crystal (eg from the Cry1Ac expressing strain HD73), solubilize it at pH10.5 (say) and then dialyse in a buffer at pH8 the protein will remain soluble. This is true of most Cry proteins. What is certainly not true is that these Bt crystals will solubilize at pH 8, however I can see no evidence that this is true for the Cry14Aa crystals either. All solubilization experiments in this manuscript appear to have been performed at pH10.5

3) The above claim might therefore just refer to the fact that some Cry14Aa was found in a soluble form in the sporulated/sporulating Bt culture which was grown at pH8. Although the authors state that this is not the case for other Cry toxins they do not provide evidence or citations to support this. The toxin will probably be produced in a soluble form within the bacterium before being laid down as a crystal and it is quite conceivable to imagine that some of the toxin will not be incorporated into the crystal. To claim that this is specific to Cry14Ab the authors should provide more evidence that such a situation is not seen with other Bt strains. Possibly related to this is the observation that 30ml cultures had activity whereas 1-2 ml ones didn't. This needs further explanation / experimentation.

We have changed the manuscript to make it clearer that in addition to the presence of crystals that were not soluble at pH 8, there was also a significant amount of soluble protein produced by the native strain and by *B.t.* transformed with a plasmid expressing the Cry14Ab gene. We discovered and characterized the activity by fractionating culture supernatants of the native strain, without solubilizing the crystals. We have removed the claim that this is an unusual property of Cry14Ab since we don't have data on this point for other Cry toxins. Specifically, we changed lines 150-152, 189, 223-224, and 459-462.

4) From the description of the methodology I presume that the authors were screening supernatants of Bt strains looking for novel virulence factors and came across the toxin through this route. Presumably Cry14Ab is also found in the crystal of the strain being characterized? This should be made clear.

We have changed lines 89-99 and 150 to make it clearer that we were in fact screening culture supernatants, and found one that was active against *C. elegans*. We did not make use of the crystals from the native *B.t.* strain when we identified and characterized the protein. However, as pointed out in lines 169-171, when we expressed the Cry14Ab gene heterologously in an acrySTALLIFEROUS strain of *B.t.* we saw crystals that were visually similar to the crystals produced by the native strain, and when we solubilized those crystals at high pH the protein was active.

5) The data provided in the manuscript certainly indicate that Cry14Ab is not proteolytically processed at the C-terminus by the range of proteases used. As the authors point out the conditions in the nematode gut may provide an environment where cleavage can happen. Having presented all the protease resistance data the reader will need to be told what happens to the protein (proteolytically speaking) within the gut. I suspect that the authors attempted such an experiment but without getting clear data and would suggest that either clear data are obtained, or this section removed.

The reviewer is correct that we did not obtain clear results from experiments in which we attempted to determine if *C. elegans* cleaves Cry14Ab to a stable core. We included the data on the fact that a wide variety of proteases did not cleave the protein to a stable core in vitro because in our experience this is unusual for Cry proteins. The result fits with the fact that the Cry14Ab amino acid sequence doesn't contain any obvious protease recognition sites in the location where Cry proteins tend to be cleaved to remove the C-terminal half of the protein. For this reason we prefer to retain this section even though we have not been able to tie it to the behavior of the protein in vivo. However, if the reviewer still feels strongly that we should remove this section we will do so.

6) Line 65. When referring to toxins that “only affect nematodes that feed on those plants” the authors should clarify the point that I believe that they are making - that the protein may affect other nematodes but will not have an effect in the field since those nematodes will not be exposed to the toxin.

We deleted the sentence in lines 63-65 since we don't wish to imply anything about environmental safety, which is a topic not covered in this manuscript.

7) Fig1. Box d is meant to show sporulated Bt yet I can't see any spores.

We changed panel d) in the figure to a field that shows both spores and crystals.

8) Fig3. I would prefer that the X-axes of boxes c and d have an identical range

We have made this change (lines 204-205).

9) Fig3. The graph itself (rather than just the legend) should state the source of the protein

We have made this change (lines 204-205).

10) Line 270 spelling of trypsin

We have changed line 270.

11) Line 463 you can't say that it doesn't have activity against other tested insects without naming those insects.

We have deleted lines 462-464, because, as mentioned above, we don't wish to imply anything about environmental safety, which is a topic not covered in this manuscript.

12) 12 Line 512 spelling of grown

We have changed line 512.

13) References – lots of italicization missing

14) Ref 16 name?

We have fixed several technical errors in the references.

Reviewer 2.

(1) Previous size exclusion experiments have shown that the feeding tube produced by cyst nematodes can only take up molecules approx. 30 kDa and smaller (Goverse et al., 1998; Bockenhoff et al., 1994; Urwin et al., 1997), but this paper states that a 132 kDa (Cry14Ab) protein is able to enter into the nematode gut. A possible reason could be that mechanical damage caused by nematodes inside roots activates a novel set of proteases that cleave Cry14Ab to a protease-resistant core, which is in turn ingested by nematodes. Considering environmental concerns related to use of Bt, such a possibility should be tested and discussed in the manuscript.

Follow up question from reviewer: I would like to know both (a) whether *H. glycines* are ingesting full-length protein versus truncated protein, and (b) whether full-length protein might be environmentally safer than truncated protein or vice versa. Authors claim that most Cry proteins currently in use in transgenic crops are in fact expressed as genetically truncated protease-resistant cores rather than as full-length proteins. But where do I find this information in manuscript? Why not present such an information/possibilities (and data if any) and discuss it faithfully?

Lines 418 to 447 have been rewritten to make it clear that full-length protein (132 kDa) can be ingested by J2 nematodes *in vitro* in a liquid assay using octopamine as a feeding stimulant, but that we haven't established if that is happening *in vivo* in nematodes feeding on roots. We make no statement in the manuscript about the relative environmental safety of truncated versus full-length Cry proteins. We have removed one sentence in lines 63-65 so that it won't appear that we are implying anything about environmental safety, which is not a topic of this manuscript.

(2) Fig. 3: Important experimental details have been omitted, making it difficult to judge the rigor and reliability of the data. What do the error bars indicate? Why do some data points have them but others not? How many technical and biological replicates were performed? What is the statistical significance of the data? Please provide all such information in the figure legend. Also, note that bioassay is misspelled and that the graphs are of suboptimal quality, with the y-axis less a legend than a title.

We have modified Figure 3 (now Figure 4) and its legend to make the changes and to give the information the reviewer requested. These experiments were only done once, so we make it clear that the EC_{50} values are very approximate.

(3) Fig. 5: What is the statistical relevance of the data? What do the error bars represent? Why do only some data points have error bars? Why is there a difference between truncated unlabeled and truncated labeled? At concentration 0 there seems to be two samples for truncated unlabeled (diamonds). I suggest using a colored graph to better define the data points. Also, why were all four proteins not tested at the various concentrations?

Figure 5 (now Figure 6) has been modified to remove erroneous data points at 0 concentration, and to use color symbols. The legend has been modified to indicate the number of replicates. The text starting at line 239 and starting at line 250 has been modified for greater clarity and to make the point that the experiments were only intended to insure the cleaved and/or fluorescently labeled protein was still active. We did not try to determine the relative activity precisely. The reason all four proteins weren't tested at all the concentrations is because the experiments were done at different times and were only intended to confirm that activity had not been lost.

(4) Fig. 9: Consider splitting the graphs into F1-30 day, F1-60 day, and F2-60 day or perhaps just simplifying by using only one time point. Second time point can go into supplementary data. How many plants were used (n =?)? Again, explain the error bars.

Figure 9 (now Figure 10) has been remade in a way that we believe is simpler and clearer, and the number of plants used is now stated. The legend has been rewritten to give statistical information.

(5) Fig. 10: Crucial details are missing in the figure legends. Also, how many plants were tested (n =?).

Figure 10 (now Figure 11) has been remade and the legend has been rewritten to give more details and statistical information. The text that refers to the figure has been rewritten (starting at line 389).

(6) Even though the materials and methods section discusses methods of identifying and isolating the bacterial strain, more detail is needed on how to do so (lines 499–506). How was the strain actually isolated? Data from worm bioassays should be added to this section to support these findings.

Follow up question from reviewer: (6) This is pertaining to “Discovery of strain” in the results section. Authors wrote that “A bacterial strain isolated from soil was identified that showed activity against *C. elegans* in a liquid bioassay”. But what kind of soil? How were the bacterial strain/s isolated from soil? I suggest to provide detailed information in methods section. Similarly, I would also like to see data/images for liquid bioassays. As it is written, this section describes a set of observations but without any data.

A paragraph has been added at the beginning of the Materials and Methods section (line 483) describing how strains were isolated from soil. We took panel a) out of Figure 3 and turned it into Figure 1 so that the bioassay scoring system is explained at the beginning of the Results section, and we added a paragraph about the bioassay (line 89). We have also described the bioassay method more clearly in the Materials and Methods section (lines 489-496), and have pointed readers to what is now Figure 1, showing examples of the visual scoring system.

(7) Many abbreviations were used throughout the manuscript without clearly establishing their meanings. Readers are likely to be confused at being left to search for abbreviations’ meanings themselves (cf. line 135).

We have spelled out all abbreviations. Specifically, we changed lines 41, 70, 109, 135, 138, 143, 154, 235, 236, 250, 314, 350, and 505.

(8) Lines 615–638, the field trial description, currently included in the materials and methods section, should perhaps also be included in the results section.

Both the body of the paper (starting at line 389) and the materials and methods section (starting at line 616) have been rewritten as suggested.

(9) The word bioassays is used multiple times without citation and without explanation of how bioassays were performed (lines 101–116).

The *C. elegans* bioassay method is now described more clearly in the Materials and Methods section (lines 484-496), and at the beginning of the results section (starting at line 89). We have also taken panel a) out of Figure 3, and made it into a new figure (it is now Figure 1) so that the visual scoring system for the bioassays will be presented at the start of the results section.

(10) A number of key observations and conclusions are arrived at abruptly without proper citations (cf. lines 144–145, 150–151). A thorough revision of literature is recommended.

Throughout the manuscript we have added citations for observations and conclusions. Some examples include lines 51, 60, 63, 74, 89, 148, 152, 161, 306, 310, 345.

(1) Fig. 1: Please provide scale bars for all microscopic images. Also, it is unclear exactly what these figures show? What do a, b, c and d indicate? Please label crystals in each image.

Figure 1 (now Figure 2) and its legend have been modified for greater clarity and to add a scale bar, as requested. Scale bars have also been added to all the other figures that contain micrographs.

(2) Fig. 4: Using different arrangements of proteases and ladder in three different gels may confuse readers.

The position of the molecular weight markers was changed in each gel to avoid mixing up the gels while they were being run. The purpose of the figure is to show that none of the proteases digested Cry14Ab to a stable core, so all the lanes look very similar and their order doesn't matter much (except for a couple of lanes where the Cry14Ab protein was completely destroyed).

(3) Figs. 6 and 7: Please add scale bars. Also, I suggest placing figure 7 earlier in the manuscript.

We have added scale bars to all figures that contain micrographs. We prefer to keep Figure 7 (now Figure 8) in its current location because it is the first image of *H. glycines*, while all previous images are of *C. elegans*. We have modified line 290 to make it clearer that at this point in the manuscript we are changing our focus from *C. elegans* to *H. glycines*.

From the editor: * Please replace your bar graphs with plots that feature information about the distribution of the underlying data. All data points should be shown for plots with a sample size less than 10. For larger sample sizes, please consider box-and-whisker or violin plots as alternatives. Measures of centrality, dispersion and/or error bars should be plotted and described in the figure legend.

Figure 10 (now Figure 11), its legend, and the text that refers to the figure, have all been changed as requested.

REVIEWER COMMENTS

Reviewer #1 (Remarks to the Author):

I am pleased to note that the authors have made a significant effort to address my previous concerns and the manuscript is greatly improved as a result. I have a few remaining comments but these are relatively minor in nature. In my opinion the conclusions drawn from the work are of interest and supported by the data.

- 1) I would still question the use of the word novel (title, abstract, line 70). This is a new protein but one that is highly similar to existing ones. I share the previously noted opinion (<https://www.nature.com/articles/43049>) that novel should be restricted to very different situations, for example a nematode-active protein with a structure different to those previously described for such proteins. In the end this should be an editorial decision.
- 2) Much greater clarity has been provided on the properties of Cry14Ab. I believe that the manuscript is still missing a line stating that it is believed that Cry14Ab is found in the crystal of the native strain. Although this is implied it is never stated, or supporting data shown.
- 3) Related to the above, for Figure 4a it should be stated how the relative concentrations compare between the whole culture and the supernatant. The readers will assume that they are equivalent ie a value of 1 represents undiluted whole culture and the supernatant from undiluted whole culture. This should be clarified, or if one of the samples has been diluted (for the "1" sample) this should be stated. I note that there is no obvious data point for the s/n at value 1 – is there a reason for this?
- 4) For figure 6 the concentrations should be converted to molar (nM) to allow a better comparison. 1ug of truncated protein will contain twice as many molecules as 1ug of full length. This conversion may bring the full length and truncated unlabelled protein a bit closer and provide stronger evidence that the C-terminus has no significant effect.
- 5) Sorry for being a pedant but on line 593 DNA is not transformed into a bacterium, a bacterium is transformed with DNA. DNA can be introduced into a bacterium by transformation.

Reviewer #2 (Remarks to the Author):

Authors have addressed most of my remarks on last version of the manuscript. Data seems reliable and message is clear that Cry14Ab have the potential to control SCN population in green house as well as in field conditions. A remaining issue is not having enough number of biological replicates (Fig 3, 4, 6, 10, 11) for some of the major experiments. However, I am not a statistician and I will leave it to the Editor to decide whether it meets their publication criteria. I should mention that authors did use multiple independent transgenic lines in figure 10, which strengthens their claim that transgenic lines are resistant to nematodes. A minor issue is figure 6 where each data point represents average of two technical replicates but error bar is missing for majority of the data points. Was there no variation in data?

Responses to second round of reviewers' comments, manuscript NCOMMS-20-46371A, March 2021. Our responses are in blue.

Reviewer #1 (Remarks to the Author):

I am pleased to note that the authors have made a significant effort to address my previous concerns and the manuscript is greatly improved as a result. I have a few remaining comments but these are relatively minor in nature. In my opinion the conclusions drawn from the work are of interest and supported by the data.

1) I would still question the use of the word novel (title, abstract, line 70). This is a new protein but one that is highly similar to existing ones. I share the previously noted opinion (<https://www.nature.com/articles/43049>) that novel should be restricted to very different situations, for example a nematode-active protein with a structure different to those previously described for such proteins. In the end this should be an editorial decision.

We have changed the word novel to the word new in lines 2, 27, and 70.

2) Much greater clarity has been provided on the properties of Cry14Ab. I believe that the manuscript is still missing a line stating that it is believed that Cry14Ab is found in the crystal of the native strain. Although this is implied it is never stated, or supporting data shown.

In line 174 we previously indicated that when working on the native *B.t.* strain we only made use of the soluble form of Cry14Ab and not the crystals to purify and identify the protein. Also, in lines 192-3 we previously indicated that when the protein was produced heterologously in an acrySTALLIFEROUS strain of *B.t.* it produced crystals similar in appearance to the ones produced by the native strain. We have now added a statement in line 194 indicating that the similarity in appearance leads us to suspect that the crystals produced by the native strain contain Cry14Ab, although we did not analyze those crystals.

3) Related to the above, for Figure 4a it should be stated how the relative concentrations compare between the whole culture and the supernatant. The readers will assume that they are equivalent i.e. a value of 1 represents undiluted whole culture and the supernatant from undiluted whole culture. This should be clarified, or if one of the samples has been diluted (for the "1" sample) this should be stated. I note that there is no obvious data point for the s/n at value 1 – is there a reason for this?

In line 222 (the legend for Figure 4) we have now stated that a relative concentration of 1 indicates undiluted samples. The symbols have been jittered in the horizontal direction to make superimposed symbols more visible.

4) For figure 6 the concentrations should be converted to molar (nM) to allow a better comparison. 1ug of truncated protein will contain twice as many molecules as 1ug of full length. This conversion may bring the full length and truncated unlabelled protein a bit closer and provide stronger evidence that the C-terminus has no significant effect.

Figure 6 has been revised as requested. In addition, we now show both technical replicates for each concentration in the figure, rather than showing the average. The symbols have been jittered in the horizontal direction to make superimposed symbols more visible.

5) Sorry for being a pedant but on line 593 DNA is not transformed into a bacterium, a bacterium is transformed with DNA. DNA can be introduced into a bacterium by transformation.

We have changed lines 343, 570, 593, 647, 658, 659, and 661 to state that bacteria and plants were transformed with DNA constructs.

Reviewer #2 (Remarks to the Author):

Authors have addressed most of my remarks on last version of the manuscript. Data seems reliable and message is clear that Cry14Ab have the potential to control SCN population in green house as well as in field conditions. A remaining issue is not having enough number of biological replicates (Fig 3, 4, 6, 10, 11) for some of the major experiments. However, I am not a statistician and I will leave it to the Editor to decide whether it meets their publication criteria. I should mention that authors did use multiple independent transgenic lines in figure 10, which strengthens their claim that transgenic lines are resistant to nematodes. A minor issue is figure 6 where each data point represents average of two technical replicates but error bar is missing for majority of the data points. Was there no variation in data?

Figures 4 and 6 have been revised to show every technical replicate for each concentration, rather than showing averages. In many cases the scores were the same for replicates, so we have jittered the symbols to make them easier to see. The EC₅₀ values derived from the data in Figure 4 are very approximate, and we do not mean to imply that a statistically significant comparison can be made between our values and values for other proteins reported in the literature. The experiments shown in Figure 6 are only intended to show that proteolytic truncation or fluorescent labeling of the protein did not abolish the toxicity of the protein towards *C. elegans*. We do not draw any conclusions in the manuscript about the exact level of toxicity. (The reviewer presumably didn't intend to refer to Figure 3, which doesn't contain a graph.)

The paragraph at lines 371-385, which is associated with Figure 10, has been revised to make it clearer that the number of replicates used in our greenhouse studies was consistent with standard published methods for these types of studies (Niblack, T. L. *et al.* A Standard Greenhouse Method for Assessing Soybean Cyst Nematode Resistance in Soybean: SCE08 (Standardized Cyst Evaluation 2008). *Plant Health Progress* **10**, 33 (2009)). The Standard Cyst Evaluation 2008 (SCE08) lays out best practices for conducting variety evaluations for resistance to soybean cyst nematode (SCN). The method recommends at least 3 replications per comparison being made, with a minimum of nine replications per entry. In figure 10 we show five comparisons being made: (a) T1 plants infested with the OP50 SCN colony for 30 days; (b) T1 plants infested with the OP50 SCN colony for 60 days; (c) T2 plants infested with the OP50 SCN colony for 60 days; (d) T1 plants infested with an HG type 2.5.7 SCN colony for 30 days; and (e) T2 plants infested with an HG type 2.5.7 SCN colony for 60 days. In all comparisons a minimum of three replicates were utilized for each entry. A power analysis of representative greenhouse SCN data suggested a minimum of five replications would be required to obtain 80% power to detect population differences with at least half the effect size of the native SCN resistance gene *rhg1b*.

Similarly, the paragraph at lines 431-447, which is associated with Figure 11, has been revised and a citation has been added to make it clearer that the number of replicates used in our field trial exceeds the requirements of standard published methods (Wang, J. *et al.* Soybean Cyst Nematode Reproduction in the North Central United States. *Plant Disease* **84**, 77–82 (2000)). The field experiment utilized four replications in each of two years (2010 and 2011) for a total of eight replications. This follows the standard replication for field SCN studies, which utilize a minimum of three replications in each of two years, for a total of six replications. A power analysis of representative field SCN reproduction data suggested a minimum of seven replications would be required to obtain 80% power to detect population differences with at least half the effect size of the native SCN resistance gene *rhg1b*.

REVIEWERS' COMMENTS

Reviewer #1 (Remarks to the Author):

All my previous concerns/comments have been satisfactorily addressed. I am happy for this manuscript to be published without further amendments.

Reviewer #2 (Remarks to the Author):

Authors have adequately addressed my concerns and I recommend this work for publication.